# Exploration of Global Brand Value Announcements and Market Reaction

**Khuram Shafi** *,† 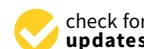**, Zartashia Hameed** †**, Usama Qadri** † **and Samina Nawab**

Institute of Information Technology, COMSATS University Islamabad, Wah Cantt 47040, Pakistan;
zartashiahameed@gmail.com (Z.H.); uxamaqadri@gmail.com (U.Q.); nawabsamina@yahoo.com (S.N.)
* Correspondence: drkhuramshafi@ciitwah.edu.pk; Tel.: +92-333-533-8904
† These authors contributed equally to this work.

**Abstract:** Brand value is an intangible asset of all firms and plays an important role in a firm's performance. Many independent firms publish the brand values of the different leading firms worldwide. Here a very simple and basic question is raised; should stockholders and investors consider and analyze brand value when they invest or not. The main objective of this study is to consider this basic question. To answer this question we considered the Global top firms in the period from September 2009 to October 2014. Results are positively significant concerning signaling theory and, it is concluded, in the context of signaling theory, that famous value brands have very important marketplace signals that can help to improve information asymmetry. Investors and stockholders can use this information regarding their investment.

**Keywords:** Interbrand; stock market reactions; event study; brand value; signaling theory

## 1. Introduction

In September 2013, when Apple Inc. revealed the iPhone 5S and 5C, their shares took a fall and decreased 2.28%, as the new devices failed to motivate investors. After one day of disclosure, shares showed another decline by 5.44% (Economic Times 2016). In September 2007, when Apple Inc. declared that the price of the iPhone will be reduced by $200, this prompted a stock value drop of 5% (Information Week 2007).

The facts disclosed above indicate that stockholders respond rapidly; rewarding corporations with an increased share price as information considered "good news" becomes available, and vice versa. A similar situation should be expected for news concerning brand values. There are numerous autonomous firms such as Equitrend, Millward Brown, Brand Finance, and Interbrand which publish yearly brand values of the leading firms and show the relative increase or decline in their value. Be that as it may, the question to be asked is, do the investors and stockholders value the publication of analysis from outside bodies when setting their value priorities? Brand value is a customer-oriented asset, and once a company has created a strong brand, any news regarding that company and brand highly affects it marketability (i.e., its stocks). Some examples are now detailed.

In September 2016, Samsung's stocks plunged 7% with the news that Samsung had stopped the manufacture of a fire-prone edition of the Galaxy Note series (CNN Tech 2016). According to an article from the Economic Times, published in 2016, when Apple Inc. unveiled the iPhone 5S and 5C in September 2013, company shares took a hit and declined by 2.28% as these new gadgets failed to successfully motivate investors. A day after the unveiling, shares showed a further 5.44% drop.

In September 2007, when Apple Inc. announced that the price of the iPhone was to be decreased by $200, this event led to a stock price drop of 5% (Information Week 2007). In July 2006, with the closing of Wal-Mart's operations in Germany, the stock price of Wal-Mart showed a 1% gain (Reuters 2006).

In August 2005, when Dell's customer satisfaction rating dipped by 6.3% (i.e., fell to 74 out of 100), its stocks, previously recorded as $49.79, closed at $36.58 (Di Carlo 2005). These cases demonstrate that stockholders respond rapidly, rewarding corporations with an increased share price as information considered "good news" becomes available, and vice versa.

Considering the abovementioned scenario, we identified the following research question: do the brand value announcements impact the stock market performance of the respective valued brand? Thus, this study investigates the publication dates of these brand value reports as events to see whether brand value news impact the stock market during the individual event windows.

## 2. Literature Review

The review of studies on the impact of marketing events on markets disclosed that these studies are mostly limited to product line or name changes.

With regard to product lines, major studies found that new product announcements have no effect on monthly stock returns, but that monthly returns are not adequately accurate to detect stock price changes, therefore making it impossible to conclusively state that investors are not quick to respond to individual new product announcements (Eddy and Saunders 1980). Afterward, it was revealed that the introduction of a new product was linked to positive returns (0.75%) and these effects were diverse across industries. The effect varied determinedly with the degree of risk and the number of announcements made, but the firm size was not related to surplus returns (Chaney et al. 1993; Chaney et al. 1991).

While analyzing brand extension, Lane and Jacobson (1995) found that the impact of brand leveraging depends on brand attitude and familiarity, but not firm size or the return on investment. The market responded most favorably to brand extensions of high esteem, high familiarity brands. The paper further concluded that investors expect the negative financial consequences of brand extension to outweigh the potential positive gains. Similarly, Hendricks and Singhal (1997) studied product delay and showed that the stock market reacted very negatively to announcements of product delay (−5.25%) and that diversified firms suffered less than focused firms. The paper further concluded that estimations of the expected delay resulted in a less negative impact than not providing an estimate at all.

Prasad Mishra and Bhabra (2001) showed that stock markets responded positively to credible new product pre-announcements (0.44%) and ignored announcements if they lacked sufficient tangible evidence. Similarly, bluffs or easily reversible announcements were ignored. Studies on product news also include research that showed that only pre-announcements of new products had a significant effect, (4.3%) Koku et al. (1997). Another author argued that new product event studies must differentiate between announcements and pre-announcements as only pre-announcements have a significant effect; (Lee et al. 2000) argued that at the time of new product introductions, first-movers experienced a positive effect (2.71%), but after early imitation, first-movers might also experience a negative reaction. Further, Chu et al. (2005), while re-examining the work of Pruitt and Peterson (1985) and Pruitt and Peterson (1986), found that new products had a positive impact on the announcing firms (0.38%), and that competitors of the firms that were announcing new products also experienced a small, depressing wealth effect. They further concluded that competitor's wealth effects were more favorable when the products introduced were innovative or unique.

Lei et al. (2013) while discussing the link between branding and innovation found that stock market response to new product announcements is negatively related to branding capabilities. The study argued that the potential reason for this is that the more famous the brand is, the higher the expectations the investors would hold with its new products. Horsky and Swyngedouw (1987) analyzed that name changes are associated with enhanced firm performance (0.61%) as name changes signal to the market that calculations to improve the firm's performance will be undertaken seriously by management.

Afterward, Bosch and Hirschey (1989) indicated a positive but statistically insignificant impact is found during the name change period (0.33%). They argued that these effects were cancelled by negative post-announcement effects. However, for firms that had previously undergone a major restructuring, the announcement of a name change was large and positive. Concerning name change events, notable studies include (Howe 1982) who proposed that company name changes appear to be financially neutral events. However, (Karpoff and Rankine 1994) deeply analyzed the work of Horsky and Swyngedouw (1987) and argued that their data was biased. Karpoff and Rankine showed with their results that reactions to name change announcements were found to be insignificant, positive, and very weak, in addition to being sensitive to sample selection and selection of the event date. Hence, their concluding remarks stated that corporate name changes may serve useful purposes, but such purposes have small valuation effects or tend to be anticipated by investors. Additional studies on name change events include that of Cooper et al. (2001), who establish that companies that change their name to a "dot.com" name gain a large and permanent increase in value, irrespective of the level of their involvement with the Internet. In the same manner, Lee (2001) concluded that dot.com name changes were linked with extensive increases in stock prices and trading activity.

Later, Kilic and Dursun (2006) found that the majority of name changes are viewed positively by the market. However, name changes by industrial goods companies with monolithic identities reduced shareholders' wealth significantly, whereas name changes by consumer goods companies with a branded identity did not affect a firm's value. Investor response was immense when other strategic investments were involved. As a contrast, cosmetic image-only name changes resulted in smaller increases than strategic name changes.

The review of different studies on the effect of marketing events on financial markets revealed that these studies were limited to product line or name changes, with very few studies undertaken which focused on brand value news. Therefore most of the research is done on a marketing context with seemingly little awareness of the fact that brand value is one of the main intangible resources and that researchers in finance award great significance to intangible assets.

## 3. Methods

### 3.1. Event-Study Model

Three pieces of information are required to undertake an event study—the names of stock-listed firms, the event dates relating to the announcement of interest, and the relevant stock prices. There are five fundamental phases in an event study. These are:

(1)　Identification of the event of interest,
(2)　Definition of criteria for inclusion of the event,
(3)　Calculation of normal and abnormal returns,
(4)　Estimation of the normal performance model, and
(5)　Performance of statistical and hypothesis tests.

Three types of information are required to undertake an event study:

(1)　the names of stock-recorded firms,
(2)　the event dates in connection to the declaration of intrigue, and
(3)　the significant stock costs.

3.1.1. Step I: Identification of Event of Interest

An adequate event is one that is; (1) likely to financially affect the firm; (2) unexpected by the market; and (3) gives new information to the market (McWilliams and Siegel 1997). In brand-related research, events may consist of the recall of a branded product, the primary introduction of an environmentally friendly product line, or the announcement of a company's aim to support the

Olympic Games. Every event has the potential to affect the stock price of a firm. The second issue relates to which dates to look at for stock price changes. If a product is suddenly reviewed, the window of interest is probably going to be short, for example, the day of and the day following the recall announcement. Moreover, recognizing the correct announcement release date to the general public can be complex. It was verified that the release date by looking at digital newsprint databases, for example, Factiva, Bloomberg Weekly, or Lexis-Nexis for the initial announcement to the general public.

### 3.1.2. Step II: Definition of Event Criteria

Event researches often studied different variables, such as firm size, investment amount, and industry type. Each of these requires a sound theoretical logic and reasoning for their consideration in the research (Johnston 2007). For instance, if the pharmaceutical industry launches a new medicine, the consideration is concentrated solely on the pharmaceutical business.

### 3.1.3. Step III: Calculation of Normal and Abnormal Returns

The distinction between a company's normal daily returns and the irregular returns experienced around the event date are measured to determine the effect of an event on shareholder value. This figure is attained by processing the day by day (or aggregate) abnormal returns accrued during the event window, minus the anticipated normal returns should no such event have happened. Two main ways to model normal returns are utilized: the constant mean return model and the market model (MacKinlay 1997; McWilliams and Siegel 1997; Srinivasan and Bharadwaj 2004). The constant mean return model is based on the idea that the mean return of a given stock is constant over time. The market model expects a linear relationship between the return of the general market portfolio and the individual stock's return. Calculation of the market portfolio is often based on a main broad-based stock index, such as the S&P 500 file (Voss and Mohan 2016), the CRSP value-weighted index, or the CRSP equal-weighted index (Srinivasan and Bharadwaj 2004). The market model is perceived as giving a more prominent ability to recognize event impacts (MacKinlay 1997; Srinivasan and Bharadwaj 2004).

### 3.1.4. Step IV: Estimation of Normal Performance Model

While the event window used to calculate the abnormal returns places emphasis on the days when information associated with the event is ready to release, the estimation window used to measure the normal performance model concentrates on "normal" trading days, generally a period well in advance of the time of information about the event being released. Normally, estimation windows are vast (around 250–600 stock exchange trading days) and are isolated from the event window by countless days (45–90) (Johnston 2007).

### 3.1.5. Step V: Statistical Calculation and Hypothesis Testing

Having decided the parameters for assessing the normal performance model, the abnormal returns are calculated and verified for significance. To investigate the data further, abnormal returns can be totaled after some time for an individual stock, and furthermore crosswise over firms (Srinivasan and Bharadwaj 2004). Where abnormal returns are especially sensational, the dollar effect or net present value can be computed to show the practical significance of the results (Chehab et al. 2016; Cooper et al. 2001; Pruitt et al. 2004; Lei et al. 2013). Test statistics in event researches are very sensitive to outliers. The effect of any one company's profits on the sample statistic can be magnified, especially when the study depends on a single event sample (Johnston 2007).

By consideration of the above criteria, the event utilized in this research is published in the Interbrand list, with the objective of showing whether the stock market responds to brand value

announcements. For this reason, Lei et al., 2013 proposed the Cumulative Abnormal Return (*CAR*) testing, that is utilized as per the following:

$$Z = \frac{1}{N} \sum_{i=1}^{N} SCAR_i \tag{1}$$

where:

$$SCAR_i = \frac{CAR_i}{(\sigma_i)\left(\sqrt{T}\right)} \tag{2}$$

So, the Equation (2) will become:

$$Z = \frac{1}{N} \sum_{i=1}^{N} \frac{CAR_i(t_1 t_2)}{(\sigma_i)\left(\sqrt{T}\right)} \tag{3}$$

where $N$ is the number of firms in the pooled sample and $T = t_1 - t_2 + 1$.

CAR is the sum of abnormal returns over the period of intervals and is shown as:

$$CAR_i(t_1 t_2) = AR_{it_1} + \cdots + AR_{it_2} = \sum_{t=t_1}^{t_2} AR_{it} \tag{4}$$

However, for the abnormal return calculation for each individual stock, we will follow the study of (Cooper et al. 2001) and compute the market adjusted returns as the abnormal returns earned by each firm. That is:

$$AR_{it} = R_{it} - R_{Mt} \tag{5}$$

The event study window for our testing will be $(-3, +3)$, $(-2, +2)$ and $(-1, +1)$, as suggested by (Lei et al. 2013).

In this event-study model, individual stock returns of the selected firms are calculated around the event dates (i.e., the publication of the Interbrand Report with the event window of $(+3, -3)$, $(+2, -2)$, and $(+1, -1)$), as mentioned above, and the event dates are obtained from Bloomberg Business Week. For the later regression equation, brand value and brand value change for each respective brand are obtained from the Interbrand Report of that respective year. The time span of the study is 7 years (i.e., from September 2009 to October 2016).

### 3.2. Hypotheses

According to MacKinlay (1997), the first published event study was undertaken by Dolley (1933) who examined the price effects of stock splits. Despite this early interest, event-study analysis failed to capture the imagination of researchers until a seminal article on the adjustment of stock prices to new information was published (Fama et al. 1969). Over the next decade, interest in event-study methodology was primarily in the domain of accounting and finance, where researchers investigated firm-specific events such as mergers and acquisitions, and broader macroeconomic effects such as the trade deficit (MacKinlay 1997).

Subsequently, other fields such as law, economics, and management embraced the methodology, focusing on issues of relevance to their specific fields such as legal liability (Mitchell and Netter 1994), the effect of the Chernobyl crisis on electric-utility stock prices (Fields and Janjigian 1989), and the departure of non-senior managers from investment banks (Bendeck and Waller 1999). From the 1980s onward, marketing researchers began to use the methodology, focusing initially on the stock price impact of new product announcements (Eddy and Saunders 1980) and deceptive advertising (Peltzman 1981). Two developments aided the expansion and dissemination of event-study methodology over the past 25 years. First, was the spread of computing and technology generally (Green et al. 2004). Second, and allied to this technological revolution, was the creation of large stock

price databases in the early 1990s. This provided researchers with relatively easy access to secondary data. Since then, daily financial data have become readily accessible to researchers through databases and software programs that provide a reasonably straightforward means of undertaking the statistical analyses involved in an event study.

Even though no significant previous literature has considered the Interbrand Report publication as an event, the previous studies on the financial effect of marketing events (Cooper et al. 2001; Prasad Mishra and Bhabra 2001; Lei et al. 2013; Yeung and Ennew 2001) have explained a specific hypothesis-testing mechanism which, with the help of, we have tested the following hypotheses for our model analysis:

**Hypothesis 1.** *Top 25 selected firms do not generate Abnormal Returns during the Event Window.*

The main hypothesis has been divided into the two categories, as suggested by the signaling theory, that should yield a negative return and a positive return on the announcement date (Lei et al. 2013). Upon the announcement of the brand value reaction in both ways can occur, so the methodology also supports this aspect. As the abnormal returns are negative and positive, we should test the same way, using the signed signal rank test (positive or negative). As we must use the robustness of the event window, it is hypothesized that:

$Alternative_{1A}$: Top 25 selected firms generate Negative Abnormal Returns during the Event Window.

$Alternative_{1B}$: Top 25 selected firms generate Positive Abnormal Returns during the Event Window.

### 3.3. Data and Samples

Brand value data is obtained from the Interbrand Report of the Top 100 Global Brands for each respective year, stock market data is collected from Google Finance, and the event dates are obtained from Bloomberg. The time span of the study is seven years (i.e., from September 2009 to October 2016).

The sample for the study consists of the Global Top 25 branded firms (Appendix A) on the following basis:

- The firm should be single-brand as it is exceptionally hard to examine the particular impact of each brand if numerous brand firms should occur.
- The valued-brand firm should be a listed company in the stock exchange as it constrains the study regarding data of unlisted firms.
- The brand of the firm should be included in the Best Global Brands (BGB) list published by Interbrand.

### 4. Results

As explained in the methodology, we have recognized the yearly publication of the Interbrand Top Global Brands report and examine whether this distinguished event significantly influences the stock performance of the organizations for which brand is considered by Interbrand in its report as one of the Top 100 Global Brands. A *z*-statistic evaluation, as proposed by Lei et al. (2013), is used to check whether CARs created during the event window are significant or not. The *z*-statistic is as per the following:

$$Z = \frac{1}{N} \sum_{i=1}^{N} \frac{CAR_i(t_1 t_2)}{(\sigma_i)\left(\sqrt{T}\right)} \qquad (6)$$

The *z*-statistics results for seven years, taken as the observation period, for each of the individual event windows. In the table below, we can see the z-statistics results of CAR values of each of the observed years for the nine individual event windows for the top 25 datasets. The rank test, which

identifies small levels of abnormal returns, was utilized to check the robustness of the CAR value. Moreover, scholars have recommended the used of nonparametric sign and rank tests for applications that require robustness against abnormally distributed data. Past research (e.g., Fama et al. 1969) has argued that daily return distributions are more fat-tailed (exhibit very large skewness or kurtosis) than normal distributions, and therefore suggest the use of nonparametric tests. As Table 1 shows, for all events windows in 2009, there is a significant negative *z*-value, therefore, for the year 2009, the null hypothesis of Hypothesis 1, that the chosen top 25 firms do not produce irregular returns, must be rejected and the alternative hypothesis, "A", in which the main 25 chosen firms create negative abnormal returns during the event window, must be accepted. 2010 is similar to 2009, as it also indicates significant negative values and we have to reject the null hypothesis of Hypothesis 1, and accept the alternative hypothesis, "A".

For event windows in the year 2011, there is a blend of negative and positive values of z, resulting in no overall significance, meaning that for 2012, we must accept the null hypothesis of Hypothesis 1. In the year 2012, a greater proportion of the event windows give a negatively significant *z*-value, apart from the event window (−3, +1), where the value isn't significant. Overall, for the year 2012, we generally accept the alternative hypothesis, "A". In the year 2013, for all event windows there is an insignificant positive value of z, which means that for 2013 despite the fact that there is a positive abnormal return, it isn't significant, so we accept the null hypothesis of Hypothesis 1.

The year 2014, differs slightly from the other years as indicates significant positive values which means firms are creating positive abnormal returns and we reject the null hypothesis of Hypothesis 1 and must accept the alternative hypothesis, "B", in which the top 25 selected firms are generating positive abnormal returns during the event window. The year 2015 is the same as 2009 because it also demonstrates a significant negative value and we reject null Hypothesis 1 accept the alternative hypothesis, "A".

**Table 1.** *z*-Statistics (year wise global branded firms).

| Event Window | z-Statistic (2009) | z-Statistic (2010) | z-Statistic (2011) | z-Statistic (2012) | z-Statistic (2013) | z-Statistic (2014) | z-Statistic (2015) |
|---|---|---|---|---|---|---|---|
| (−3, +3) | −12.4 ** | −3.91 ** | −1.28 | −5.14 ** | +1.53 | +7.49 ** | −3.58 ** |
| (−3, +2) | −11.1 ** | −3.03 ** | −0.74 | −3.45 ** | +1.31 | +5.90 ** | −2.94 ** |
| (−3, +1) | −9.89 ** | −2.75 ** | +0.19 | −1.94 | +1.56 | +4.00 ** | −2.45 * |
| (−2, +3) | −12.1 ** | −4.18 ** | −1.43 | −5.59 ** | +1.18 | +7.77 ** | −3.76 ** |
| (−2, +2) | −10.6 ** | −3.28 ** | −0.87 | −3.82 ** | +0.92 | +6.12 ** | −3.11 ** |
| (−2, +1) | −9.37 ** | −3.03 ** | +0.15 | −2.21 * | +1.16 | +4.09 ** | −2.61 ** |
| (−1, +3) | −11.5 ** | −3.86 ** | −1.79 | −5.91 ** | +0.61 | +7.61 ** | −3.90 ** |
| (−1, +2) | −9.99 ** | −2.86 ** | −1.23 | −4.04 ** | +0.27 | +5.83 ** | −3.22 ** |
| (−1, +1) | −8.69 ** | −2.57 ** | −0.12 | −2.29 * | +0.46 | +3.55 ** | −2.73 ** |

* Significant at 5%; ** Significant at 1%.

## 5. Discussion

As explained earlier, the event we recognized during this research study is the publication of the Interbrand Report of the Top 100 Global Brands. This was because, while many previous studies had used the brand value estimates of Interbrand, no such study had considered publication of the report as an important event which, itself, might cause the abnormality in stock performance of firms whose brand is recognized in this report. The results reveal that the years 2009, 2010, 2012, and 2015 indicate significant negative values, indicating that the top 25 firms do not produce abnormal returns during the event window, while the year 2011 indicates insignificant negative and positive values. The results reveal that in the years 2014 significant positive values are indicated, which shows that the top 25 firms produce abnormal returns during the event window. There is also some confusion around event studies and stock return response modeling. While both approaches are founded on similar assumptions regarding the efficient market hypothesis, and focus on the impact of

unanticipated events on stock price, they have key differences. Event studies examine the stock price impact of a specific announcement on a given day. The nature of the event may be unique, such as the announcement of a firm's name change, or it may be a recurring announcement, such as the annual release of brand value reports. The period of interest is generally an event window that focuses on the actual day of the event, or the five to ten days immediately surrounding it, based on the anticipated time taken for the new information to be absorbed by the market. In contrast, stock response modeling looks at the long-term value implications of data that may be released monthly, quarterly, or yearly, such as changes in brand equity in relation to net earnings over time. Stock return response modeling assumes that investors have access to many sources of information regarding the firm's prospects, such as sales data, return on equity, and cash flow, as well as information about the firm's marketing strategy. Together these factors affect the future cash flows of the firm (Johnston 2007).

## 6. Conclusions

Many studies have been conducted on the Brand Value estimates of Interbrand, but no study was found which considered publication of the report as a significant event which may cause the abnormality in the stock performance of firms. In this study, we considered the publication of the most common source of brand value information, as explained by Lei et al. (2013). Results are negative and significant, with the exception of 2014 where results are positively significant, in every year for top 25 branded firms. Considering the major results of significant negative abnormal returns, the most probable explanation is the "Reverse Pygmalian Effect", which expresses that high requirements prompt low performance. At the point at which the new report arrives, a "hype" is made regarding the brand-holding firm. This hype results in a type of "bubble" around the firm stock and, as this bubble is not real and based only on expectation, at some point, it bursts and causes a dip in the stock market, particularly for those organizations included in the report. In an event in 2014, four days before publication of the yearly Interbrand Report, a cyber-attack, which put 83 million records under risk, was carried out on US financial organizations. This conflict of events caused positively-significant outcomes, as with the cyber-attack, the majority of the investors went for more secure investment (i.e., strong branded stocks) causing a sudden ascent in the stock price of our chosen firms and caused positive abnormal returns during our chosen event window.

## 7. Managerial Implications

By taking into consideration that the majority of the results show significant negative abnormal returns, the most probable explanation is the "Reverse Pygmalion Effect" which states that low performance is caused by high expectations. When the new report is due, a critical ambiguous situation is created regarding the brand-holding firm. This results in a kind of "bubble" around the firm stock, and as this bubble is imaginary and based merely on expectations, sooner or later it bursts and causes a dip in the stock market specifically for the firms included in the report. In the context of signaling theory, this study recommends that high-value brands play a part as commercial-center signals that can help to reduce information asymmetry. Investors can deduce that the organizations that possess high-value brands have the required advertising, marketing capacities, administrative sharpness, and other abilities important to guarantee long-term growth.

## 8. Limitations and Future Research

As the majority of the companies considered in this study are US based firms, even though they are globally active and performing operations in other parts of the world in addition to the US, there will be a question of generalizability concerning the results. In the case of the event-study model, the limitation is regarding the choice of event. There are many brand-related events which might affect the stock performance, e.g., product announcement, celebrity association, name change, mergers, and acquisitions, etc., but this study is limited to the publication of the Interbrand Report. Finally, in the

case of the performance model, the study is limited to just three performance indicators and has ignored many others.

An extension in the context of five-factor models can be the comparison of branded firms with non-branded firms or a comparison between the different brand value publications such as Millward Brown, Equitrend, and InterbrandIn a performance model scenario, the inclusion of other performance indicators can further enhance the scope of the study.

**Author Contributions:** K.S. is using the data analysis and Z.H. collected the data and was involved in data analysis. U.Q. and S.N. were involved in the literature review and introduction section. The conclusion is written as a collaborative effort.

**Acknowledgments:** This research study is not supported by any funded agency/organization.

**Conflicts of Interest:** The authors declare no conflict of interest.

## Appendix A

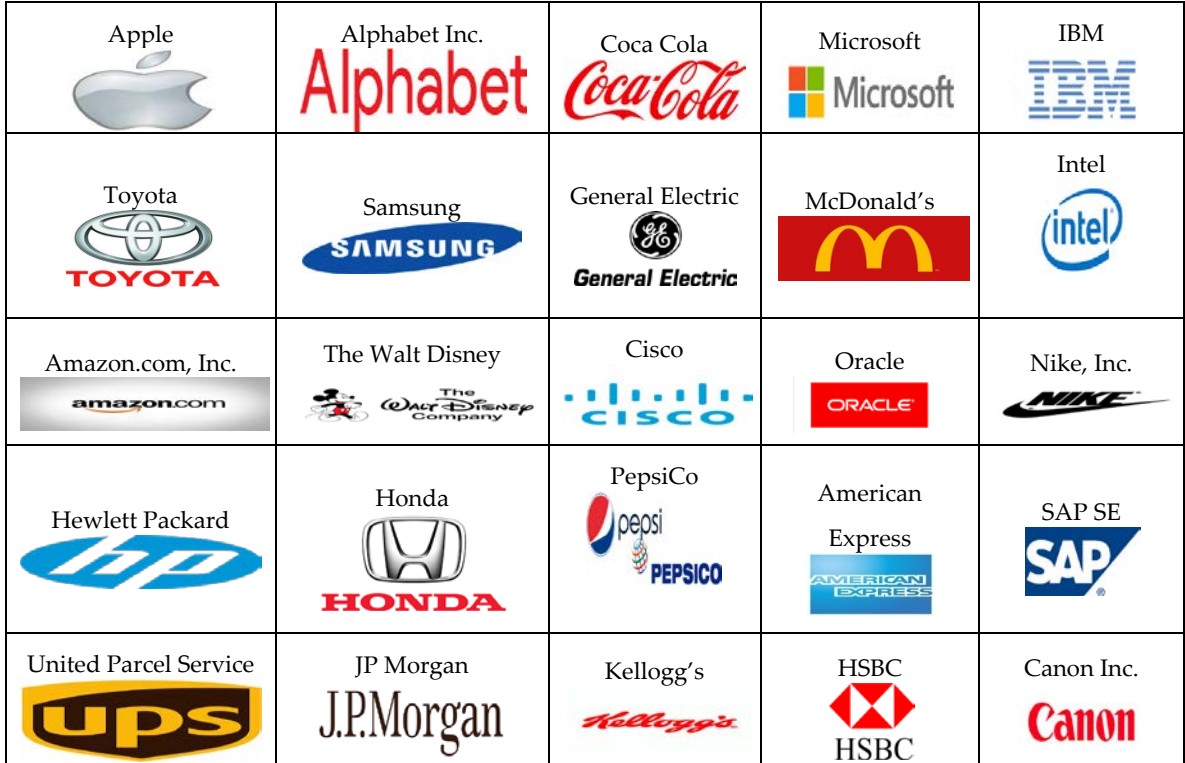

**Figure A1.** World top branded companies.

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
