# Peer review of "Exploration of Global Brand Value Announcements and Market Reaction"

_admsci, doi:10.3390/admsci8030049_

Round 1
Reviewer 1 Report
Overall the paper is scientifically sound and well written.
However, the author may add value to their work by implement a robustness test in order to validate their results.
If a different econometric technique conclude to same results, then the value of the paper will be increased.
Author Response
First of all, we would like to thank the Reviewer for giving us the opportunity to revise our manuscript, and the anonymous reviewers for taking time to review and assess this paper and provide us with many valuable comments and suggestions that contributed significantly to its improvement. We have carefully read all the comments and concerns, and have revised our paper accordingly. In this document, we respond to all the comments and suggestions made by the reviewers one by one and provide the detail of the changes made to our revised manuscript.
Reviewer: 1
Comments: Overall the paper is scientifically sound and well written.
Our Response: We appreciate the reviewer’s positive comments. We have revised the paper in view of these comments.
Comments: However, the author may add value to their work by implement a robustness test in order to validate their results.
Our Response; We have revised the paper in view of this comments. We Use Rank test to validate the result of CARS. As suggested by the Corrado and Zivney (1992) firstly and further used by many authors to validate the analysis. We have written in the opening paragraph of the result section about the robustness of the result.
Comments: If a different econometric technique conclude to same results, then the value of the paper will be increased.
Our Response; We appreciate the reviewer’s positive comments. We have highlights this econometric technique in methodology section as well its importance and why we are going to use this as appropriate to this study.
Reviewer 2 Report
The paper explores Global brand value announcements and market reaction. However, there are several weaknesses that compromise the manuscript’s potential contributions. I would like to offer the following comments as suggestions for ways in which the manuscript might be improved.
1. You go directly from your introduction section to the methods section. This is not appropriate. You need to develop a separate introduction section followed by a theoretical background section. Please note that this does not mean that you should simply split the current introduction section that you have. You have to significantly improve the introduction to clearly show the importance of the study, gaps, research questions and contributions. This should then be followed by a proper theoretical background section where you articulate the theoretical rationale underpinning the execution of your study.
2. Further to the previous point regarding the theoretical rationale: in the methodology section (specifically section 2.2) you indicate some hypotheses. These hypotheses need to be clearly developed and supported in this new theory section.
3. You need to include a discussion section to discuss your findings. In this section you should be clear how your study will contribute to the current literature and how your findings will advance current knowledge.
4. You should also add a separate section for managerial implications and another section for limitations and directions for future research.
Author Response
First of all, we would like to thank the Reviewer for giving us the opportunity to revise our manuscript, and the anonymous reviewers for taking time to review and assess this paper and provide us with many valuable comments and suggestions that contributed significantly to its improvement. We have carefully read all the comments and concerns, and have revised our paper accordingly. In this document, we respond to all the comments and suggestions made by the reviewers one by one and provide the detail of the changes made to our revised manuscript.
Reviewer: 2
Comments: The paper explores Global brand value announcements and market reaction. However, there are several weaknesses that compromise the manuscript’s potential contributions.
Our Response: We have revised the paper in view of these comments.
Comments: You go directly from your introduction section to the methods section. This is not appropriate. You need to develop a separate introduction section followed by a theoretical background section. Please note that this does not mean that you should simply split the current introduction section that you have. You have to significantly improve the introduction to clearly show the importance of the study, gaps, research questions and contributions. This should then be followed by a proper theoretical background section where you articulate the theoretical rationale underpinning the execution of your study.
Our Response: We have revised the paper in view of these comments. we have make literature section separately and introduction section as suggested in the revised document.
Comments: Further to the previous point regarding the theoretical rationale: in the methodology section (specifically section 2.2) you indicate some hypotheses. These hypotheses need to be clearly developed and supported in this new theory section.
Our Response: We appreciate the reviewer’s positive comments. we have make more theoretical support in the hypothesis section as suggested.
Comments: You need to include a discussion section to discuss your findings. In this section you should be clear how your study will contribute to the current literature and how your findings will advance current knowledge.
Our Response: We have revised the conclusion in this view.
Comments: You should also add a separate section for managerial implications and another section for limitations and directions for future research.
Our Response: We appreciate the reviewer’s positive comments. we have make different section in the revised document.
Round 2
Reviewer 2 Report
Thank you for addressing my concerns in the revised mansucript.
To improve further the manuscript I would suggest the additional chnages:
expand further the support and development of your hypotheses. Currenlty your section 3.2 is half a page, i would suggest you expand the development to at least one page.
add a new section entiteled 'discussion'. This new section should come before you conclusion section. In this new section discuss your findings and link it to the new literature review section in your paper. To give you an indication, I would expect this new section to be between 1-2 pages long.
Good luck
Author Response
Dear reviewer
Hope you are doing well. Thanks for the valuable suggestions. we have accommodate all the changes. The changes in the manuscript is in red text.
Regards
Round 3
Reviewer 2 Report
I'm satisfied with the changes.